



# Variable habitat depth of the planktonic foraminifera *Neogloboquadrina pachyderma* in the northern high latitudes explained by sea-ice and chlorophyll concentration

5  Mattia Greco[1], Lukas Jonkers[1], Kerstin Kretschmer[1], Jelle Bijma[2] and Michal Kucera[1]

[1]MARUM - Center for Marine Environmental Sciences, Leobener Str. 8, D-28359, Bremen, Germany
[2] Alfred-Wegener Institute Helmholtz Centre for Polar and Marine Research, Bremerhaven, Germany

10  *Correspondence to*: Mattia Greco (mgreco@marum.de)



**Abstract.** *Neogloboquadrina pachyderma* is the dominant species in the polar regions. In the northern high latitude ocean, it makes up more than 90% of the total planktonic foraminifera assemblages, making it the dominant pelagic calcifier and carrier of paleoceanographic proxies. To assess the reaction of this species to future climate change and to be able to interpret the paleoecological signal contained in its shells, its habitat depth must be known. Previous work showed that *N. pachyderma* in the northern polar regions has a highly variable depth habitat, ranging from the surface mixed layer to several hundreds of metres below the surface, and the origin of this variability remained unclear. In order to investigate the factors controlling the habitat depth of *N. pachyderma*, we compiled new and existing population density profiles from 104 stratified plankton tow hauls collected in the Arctic and the North Atlantic Oceans during 14 oceanographic expeditions. For each vertical profile, the Depth Habitat (DH) was calculated as the abundance-weighted mean depth of occurrence. We then tested to what degree environmental factors (mixed layer depth, sea surface temperature, sea surface salinity, Chlorophyll a concentration and sea ice concentration) and ecological factors (synchronised reproduction and daily vertical migration) can predict the observed DH variability and compared the observed DH behaviour with simulations by a numerical model predicting planktonic foraminifera distribution. Our data show that the DH of *N. pachyderma* varies between 25 m and 280 m (average ~100 m). In contrast with the model simulations, which indicate that DH is associated with the depth of chlorophyll maximum, our analysis indicates that the presence of sea-ice together with the concentration of chlorophyll at the surface have the strongest influence on the vertical habitat of this species. *N. pachyderma* occurs deeper when sea-ice and chlorophyll concentrations are low, suggesting a time transgressive response to the evolution of (near) surface conditions during the annual cycle. Since only surface parameters appear to affect the vertical habitat of *N. pachyderma,* light or light-dependant processes might influence the ecology of this species. Our results can be used to improve predictions of the response of the species to climate change and thus to refine paleoclimatic reconstructions.

## Introduction

*Neogoboquadrina pachyderma* is the most abundant planktonic foraminifera in the Arctic and its marginal seas, where it also dominates the pelagic calcite production (Schiebel et al., 2017; Volkmann, 2000). When the organism dies, its calcite shells sink to the seafloor and when preserved in the sediments, they serve as a source of information on the physical state of the ocean in the past (Eynaud, 2011; Kucera, 2007). To understand the origin of the paleoceanographic proxy signal and to predict the production of the species under varying physical conditions, including projected future change scenarios, it is important to constrain the factors that determine its vertical habitat. Previous work has shown that the seasonality of *N. pachyderma* production follows the timing of food availability, which is tightly linked with temperature (Jonkers and Kucera, 2015; Tolderlund and Bé, 1971). On the other hand, the vertical habitat of the species is variable and appears hard to predict (Xiao et al., 2014).

Previous studies proposed different abiotic factors as drivers of *N. pachyderma* vertical distribution including temperature (Carstens et al., 1997; Carstens and Wefer, 1992; X. Ding, R. Wang, H. Zhang, 2014), density stratification





(Simstich et al., 2003) and the depth of the subsurface chlorophyll maximum indicating food availability (Kohfeld and Fairbanks, 1996; Pados and Spielhagen, 2014; Volkmann, 2000). Next to environmental factors, the behaviour of the species itself, such as its ontogenetic vertical migration (Bijma et al., 1990; Erez, 1991) and day/night migration (Field, 2004), or morphologically hidden cryptic diversity (Weiner et al., 2012) , could also influence the vertical habitat observed in a single

profile. However, the Arctic and the North Atlantic are inhabited by a single *N. pachyderma* genotype (Type I) (Darling et al., 2007), indicating that the variable depth habitat of the species cannot be attributed to cryptic diversity. On the other hand, analysis of the size distribution of *N. pachyderma* shells in the Arctic by Volkmann (2000) suggested a synchronised reproduction around the full moon, with sexually mature individuals descending towards a deeper habitat to release gametes. Similarly, diel vertical migration (DVM) is known to confound observations of vertical distributions patterns of Arctic

plankton (Berge et al., 2009). Although the only study on DVM in polar waters on *N. pachyderma* showed no evidence of this phenomenon (Manno and Pavlov, 2014), it was based on observations during the midnight sun with relatively weak changes in light intensity and the existence of DVM in *N. pachyderma* during other times of the year cannot be firmly ruled out. Therefore, the influence of the two ecological patterns on the depth habitat of *N. pachyderma* has to be considered in analysis of our compilation of vertical profiles.

The lack of consensus on potential drivers of habitat variability in *N. pachyderma* calls for a systematic approach synthesizing new and existing observations into the same conceptual framework. In addition, there is now an opportunity to compare observations with predictions of a numerical model in the same framework. This opportunity arises from the recently extended model PLAFOM2.0, which can predict the seasonal and vertical habitat of *Neogloboquadrina pachyderma* (Kretschmer et al., 2018). To this end, we assembled existing vertical population density profiles of this species from the

Arctic and North Atlantic, combined these with new observations from the Baffin Bay and associated the observations with oceanographic data. Based on an analysis of the resulting dataset, we present a new concept that explains habitat depth variability in this important high-latitude marine calcifier. Next to three previously proposed environmental driver of habitat variability (temperature, stratification, food availability), we also consider chlorophyll concentration at the surface as a measure of productivity, testing for the possibility that the foraminifera are attracted to food at the surface, salinity, testing

for the possibility of the foraminifera evading low salinity surface layers, and sea-ice concentration, testing for the possibility that the foraminifera habitat respond to sea-ice related variability in light, atmospheric exchange and/or mixing.

## 2 Material and methods

Our analysis is based on a synthesis of existing and new vertical density profiles of *N. pachyderma* from the high-northern latitudes. We exclude the Pacific Ocean because it is inhabited by a distinct genetic type of *N. pachyderma* with potentially

different ecology (Darling et al., 2007). We compiled 97 population density profiles of *N. pachyderma* collected during 13 oceanographic expeditions between 1987 and 2011 (Fig.1). We excluded one profile from Jensen (1998), where the



abundance maximum occurred anomalously deep (below 500 m) and which we thus suspect to reflect an error (i.e. due to sample mislabelling). We retained all other profiles, despite the differences in the mesh size, counted size fraction and vertical resolution. The remaining compilation is representative of the Eurasian Arctic and its marginal seas, as well as of the North Atlantic, but contains no data from the oceanographically distinct Baffin Bay. To close this gap, we extended the compilation by generating new data from eight plankton tow profiles collected during the MSM09 cruise in 2008 (Fig.1). At all stations sampling was carried out down to 300 m using a multiple closing plankton net (HydroBios, Kiel) with a 50 × 50 cm opening and a 100 μm mesh (Kucera et al., 2014). The vertical distribution of planktonic foraminifera was resolved to nine levels by conducting two casts at each station (300–220 m, 220–180 m, 180–140 m, 140–100 m, 100–80 m, 80–60 m, 60–40 m, 40–20 m, 20–0 m). After collection, net residues from each depth were concentrated on board, settled and decanted, filled up with 37% formaldehyde to a concentration of 4% and buffered to pH 8.5 using pure solid hexamethylenetetramine ($C_6H_{12}N_4$) to prevent dissolution, and refrigerated. Specimens of planktonic foraminifera were picked from the wet samples under a binocular microscope and air-dried. All individuals in the fraction above 100 μm were counted and identified to species level following the classification of Hemleben et al. (1989) and Brummer and Kroon (1988). Full (cytoplasm-bearing) tests were counted separately and considered as living at the time of sampling. Counts were converted to concentration using the volume of filtered water determined from the product of towed intervals height and the net opening (0.25 $m^2$).

For the new profiles from the Baffin Bay, water temperature and salinity were measured with conductivity–temperature– depth (CTD) device deployed before each plankton tow. In order to obtain the vertical profiles of algae pigment concentrations, a submersible fluorospectrometer (bbe Moldaenke) was used for all the stations (Kucera et al., 2014). For the remaining profiles from the literature, physical oceanographic data and chlorophyll a concentrations profiles for each station were, if available, obtained from CTD profiles retrieved from PANGAEA data repository using the R package "pangaear" (Simpson and Chamberlain, 2018). Sea surface parameters, sea surface temperature (SST), sea surface salinity (SSS) and surface chlorophyll concentration, were obtained from CTD profiles and Niskin bottles by averaging all the values from the first 5 meters. The depth of the chlorophyll maximum (DCM) was determined from vertical profiles of chlorophyll concentration obtained from either water column profiles or discrete measurements from Niskin bottles. The depth of the mixed layer (MLD), defined as the depth where in situ water density varied by more than 0.03 kg/$m^3$ as in De Boyer Montegut *et al.* (2004), was calculated from the CTD profile of each station using a custom function in R. No vertically resolved profiles of environmental variables were available for plankton net hauls collected during the expeditions NEWP93, ARK-IV/3, ARK-X/1, ARK-X/2, M36/3, and M39/4. These profiles could thus only be used for the analysis of ontogenetic and diel vertical migration. In addition to the in-situ data, daily sea ice concentrations for the location of each profile were extracted from 25 × 25 km resolution passive microwave satellite raster imagery obtained from the National Snow and Ice Data Centre (Boulder, Colorado, USA) for 1979–2011 using a custom function in R (R Core Team, 2017). We used the data to determine sea-ice concentration at the time of collection and also to retrieve the time after sea-ice break for





all the stations that were sea-ice free at the moment of sampling. The date of the most recent sea-ice concentration maximum was used to retrieve the time by subtracting the days until the sampling date. Finally, the time of the collection was compared to the time of sunrise and sunset for each station determined using the R package "SunCalc" (Agafonkin and Thieurmel, 2018) to distinguish day-time and night-time collections. The sampling date was used to determine the lunar day using the R package "lunar" (Lazaridis, 2015).

The cross plots in Fig. 2 show how the final compilation of 104 profiles covers the environmental space and how the observations are spread across the seasons and the lunar cycle. The sampling is strongly biased towards the summer but the lunar cycle is completely covered. Most of the profiles were collected under midnight sun conditions, leaving only 28 profiles that could be used to test the diel vertical migration (Table 2). The profiles cover SST conditions between -2 and 7°C and contain profiles taken across the entire range of sea-ice concentrations. Since sea-ice concentration at the studied profiles was not linearly related with SST, the compilation should allow to assess the effect of the two variables independently (Fig.2c). Productivity, expressed as surface chlorophyll a concentration, is neither correlated with temperature. The most productive stations were located in the Baffin Bay and in the Fram Strait with surface chlorophyll concentrations ranging between 2 and 4 mg m$^{-3}$ (Fig. 2b). Surface salinity was mostly around 33 PSU, only in the Laptev Sea we observed values below 30 PSU.

To facilitate the analysis of habitat depth across density profiles with observations at different depth intervals, the density profiles were summarized into a single parameter, depth habitat (DH), which is the abundance-weighted mean depth calculated using the mid points of the collection intervals (Fig.3), as in Rebotim et al. (2017). Since counts of living and dead specimens were not available for all the stations, total counts were considered. However, where possible, we also derived the average living depth (ALD) to assess possible biases deriving from using total counts to constrain depth habitat. This comparison showed that ALD was highly correlated with DH and on average 11 meters shallower than DH, which thus represents a slight systematic overestimation of the actual living depth of *N. pachyderma* (Fig.4). Exceptions are stations MSM09/466, 55/84, and 36/069 where the observed ALD was deeper than DH due to the high number of dead specimens in the upper catch intervals. The appropriateness of a single parameter (DH) as an indicator of the distribution of *N. pachyderma* in the water column was further tested using a multivariate approach. We determined profile-standardized concentrations calculated for 5 depths (0-50, 50-100, 100-200, 200-300, 300-500) for all the stations and performed a principal component analysis (PCA) on the relative abundances in the sampling intervals using the R package "vegan" (Oksanen et al., 2018). The two first principle components explained 43% and 32% of the total variance in the relative abundance in the water column. The first axis exhibited negative loadings for the deeper intervals (100-200, 200-300, 300-500) and positive loadings for shallow intervals 0-50 and 50-100, indicating that it describes a depth-changing unimodal distribution (Fig 4b). Mapped on the PC1 loadings, DH showed a significant correlation (Pearson $r = -0.88$, $p$-value $<0.01$) indicating that all profiles had a single maximum and the depth distribution can be collapsed into a single variable (Fig 4b).

We start our analysis by considering the potential effect of diel vertical migration and the possibility of synchronised vertical ontogenetic migration associated with the lunar cycle. Despite its potential importance (Rebotim et al.,



2017), we cannot analyse seasonal variation in depth habitat because only a single season was sampled. The influence of diel vertical migration on DH was assessed by dividing samples in two groups based on whether they were collected during the day or during the night. The two groups were tested for homoscedasticity using an F- test and then a t-test was performed. To investigate the effects of the lunar cycle on the depth habitat of *N. pachyderma,* we used a periodic regression following the

approach described in Jonkers and Kucera (2015). In the next step, we analysed the relationship between DH and sea surface temperature, sea surface salinity, mixed layer depth, surface chlorophyll concentration, depth of chlorophyll maximum and sea-ice concentration. We use linear regression to assess if any of the variables individually predicts a significant part of the DH variability and the variables that showed significant correlation with DH were used to construct a multiple linear regression model allowing interactions. The use of linear regression assumes normality, which was tested, and linearity in

the relationship, which is assumed, but prevents overfitting and therefore all estimates of goodness of fit in our models can be considered conservative.

## 3 Results

The depth habitat values derived from the abundance profiles ranged from 26 m to 283 m with an average of 100 m (IQR=

54.95). The deepest observation comes from the Fram Strait, the shallowest from the Baffin Bay.
An independent-samples t-test revealed no evidence for an effect of diel vertical migration on the observed *N. pachyderma* vertical distribution (Table 1). Similarly, the periodic regression showed no significant effect of lunar phase on DH ($p = 0.17$, Adjusted $R^2 = 0.029$) (Table 2). In the subsequent analyses we could thus focus on abiotic factors in explaining vertical habitat variability in *N. pachyderma*. Bivariate linear regressions against DH carried out on a subset of 66 profiles for which

all of the tested environmental parameters were available yielded a significant relationship only for chlorophyll concentration at the surface (Fig. 5a). However, we noticed that profiles from stations where sea-ice was present appeared to show a relationship with sea-ice concentration and we thus carried out separate analyses for profiles with and without sea-ice. We found no significant correlation between DH and the variables SST, SSS, MLD and DCM neither the complete data set nor in the subsets (Fig.5a). Chlorophyll concentration at the surface appeared to be the only parameter showing

significant negative correlation in both the complete dataset ($r = -0.28$, $p < 0.05$) and the sea-ice free subset ($r = -0.60$, $p < 0.01$). A negative correlation between DH and sea-ice concentration was observed in the subset including ice-covered stations ($r = -0.38$, $p < 0.05$). Following the initial variable selection, where only profiles for which all variables were available were considered, we then extended the analyses to all profiles where sea-ice concentration and/or chlorophyll concentration at the surface were available. These analyses confirm the significance of the relationships (Figs. 5b and c).

30       In the Arctic, the break-up of the sea-ice is normally followed by a pulse of productivity (Leu et al., 2015), making the two tested variables potentially causally connected in a time-transgressive manner. To test for the presence of such a relationship, we tested the relationship between DH and the number of days since sea-ice break-up. To decrease the



collinearity between sea-ice and productivity, the analysis was restricted to 18 profiles from stations with chlorophyll concentrations <0.5 mg m$^{-3}$. This analysis shows that DH significantly increases with time after the sea-ice break-up ($r = 0.65$, $p < 0.01$) (Fig. 6). In the final step, we combined the three variables that individually showed significant effect on DH for at least one subset of the profiles and constructed a multiple regression model to predict the depth habitat of *N. pachyderma* based on sea-ice concentration and the interaction between chlorophyll concentration at surface and days after the sea-ice break. A linear formulation of the model is significant ($p < 0.01$) and the model explains 29 % of the depth habitat variability in *N. pachyderma* (adjusted $r^2 = 0.29$). Next, we tested a non-linear relationship, considering the log-normal nature of the DH. This model leads to a marginal improvement (adjusted $r^2 = 0.32$) (Table 3).

Finally, we evaluate how PLAFOM2.0 (Kretschmer et al., 2018) captures the observed patterns in *N. pachyderma* depth habitat. To this end, we assess the relationship between modelled DH of *N. pachyderma* and SST, SSS, MLD, DCM and chlorophyll concentration for summer months in the geographic area covered by the compilation (Fig.1). Although PLAFOM2.0 simulations also indicate a dominantly subsurface summer depth habitat of *N. pachyderma*, the modelled DH is shallower than observed, with values ranging between 9 and 127 meters (Fig.7). Contrary to observations, the modelled DH shows the strongest correlation with the depth of the mixed layer ($r = 0.57$, $p < 0.01$). Moreover, the observed relationship between the modelled DH and sea-ice and chlorophyll concentration is weak and of opposite sign to the observations (Figs.8a-b).

## 4 Discussion

Previous research indicated the absence of DVM in *N. pachyderma* in the Fram Strait (Manno and Pavlov, 2014) but the fact that the sampling was carried out during the midnight sun led the authors to concede that the species still could engage in DVM in the presence of a diurnal light cycle. Indeed, studies on copepods in the Arctic showed that natural patchiness rather than DVM is responsible for shifts in vertical distribution in periods of midnight sun, while in late summer/early autumn, when changes in the diurnal light cycle are apparent, DVM can be observed (Blachowiak-Samolyk et al., 2006; Rabindranath et al., 2011). Our compilation allowed us to assess the behaviour of *N. pachyderma* under changing light condition, revealing no evidence for DVM (Table 1). Similarly, a recent investigation on the presence of DVM in planktonic foraminifera from the tropical Atlantic found no evidence for this phenomenon in any of the analysed species (Meilland et al., 2019). Our observations thus add to the existing consensus that planktonic foraminifera are unlikely to participate in DVM. Although we cannot rule out DVM on a very small vertical or geographical scale, we conclude that the observed variability in habitat depth of *N. pachyderma* in our compilation is likely not biased by DVM, allowing us to investigate other potential drivers.

The reproduction of many species of planktonic foraminifera appears synchronized on lunar or semi-lunar cycle (Bijma et al., 1990; Jonkers et al., 2015; Rebotim et al., 2017; Schiebel et al., 1997; Spindler et al., 1979), with sexually mature individuals descending towards a deeper habitat to release their gametes (Bijma et al., 1990; Erez, 1991). Volkmann (2000) analysed size distribution of *N. pachyderma* in the Arctic and found an indication for a synchronised descent of adult



individuals below 60 m during full moon. In our analysis of 104 density profiles, including those from Volkmann (2000), we found no evidence of a systematic shift towards deeper habitat associated with lunar periodicity (Table 2). Our analysis cannot resolve whether or not the reproduction in *N. pachyderma* is synchronised nor can we rule out an irregular ontogenetic vertical migration. However, the absence of a systematic relationship between DH and lunar cyclicity in our compilation indicates that a potential ontogenetic vertical migration would likely only contribute a noise component to the DH variability.

There is general consensus that *N. pachyderma* grazes on phytoplankton and it would thus seem reasonable to assume that food availability primarily influences its vertical distribution (Bergami et al., 2009; Carstens et al., 1997; Kohfeld and Fairbanks, 1996; Pados and Spielhagen, 2014; Taylor et al., 2018; Volkmann, 2000). Surprisingly, our analysis yielded no significant correlation between the position of the subsurface chlorophyll maximum and DH. Instead, the DH of the species is always located below DCM and thus most specimens of the population do not appear to be grazing at the DCM. This observation is also in contrast with the modelled relationship of DH with environmental parameters. This is because the strong relationship between DH and MLD in the model reflects a strong link between MLD and the position of the subsurface chlorophyll maximum, as also noted by Kretschmer et al. (2018). This strong link likely results from a bias in the ocean component of the Community Earth System Model (CESM1.2) propagated in PLAFOM2.0. The CESM1.2 model is known to overestimate the mixed layer depth in the Arctic by 20 to 40 metres (Moore et al., 2013). This overestimation of the MLD in the model affects ocean biogeochemistry and the light regime experienced by the phytoplankton. Specifically, a deeper mixed layer equates to a thicker layer of nutrient depletion, lowering the DCM. Consequently, the simulated depth of the chlorophyll maximum reaches 60 to 95 meters, whereas a recent survey of vertical chlorophyll profiles in the post-bloom period (May- September) in the Arctic indicated that subsurface chlorophyll maxima occur in the top 50 meters (Ardyna et al., 2013), which is also in line with the range of DCM among the studied profiles (Fig.9). Clearly, the observed preference of *N. pachyderma* for a habitat below the DCM (Fig.9) indicates that the species may not primarily feed on fresh phytoplankton. The possibility of other species of *Neogloboquadrina* feeding on marine snow particles (hence below the DCM) has been recently suggested by Fehrenbacher et al. (2018) and a similar food source, related to degraded organic mater is thus not unlikely for *N. pachyderma*.



Among the other previously considered abiotic drivers of habitat depth in *N. pachyderma*, our analysis provides no evidence for the effect of seas-surface temperature, salinity and stratification (Fig. 4). Surface water temperature is the main controller of *N. pachyderma* abundance and it defines its geographic range (Bé and Tolderlund, 1971; Duplessy et al., 1991). Temperature could therefore also be expected to influence the vertical habitat of the species. However, we found no link with surface temperature and *N. pachyderma* depth habitat. This is probably because the temperature range sampled by our compilation remains well within the tolerance limits of the species (Žarić et al., 2005) and thus temperature does not represent a limiting factor for this species and does not affect its vertical distribution. Previous research has suggested that *N. pachyderma* may avoid low salinities and preferentially occur deeper in the water column when the surface is fresh (Volkmann, 2000, see also the discussion in Schiebel et al., 2017). Like Carstens and Wefer (1992), we did not find a significant correlation between surface salinity and DH indicating that the inferred response of *N. pachyderma* to surface layer freshening only applies to situations where the salinity reaches values below 30 PSU (below the limit covered by the observations in our compilation). Finally, geochemical analyses of sedimentary and plankton specimens were interpreted as evidence for calcification depth of the species being controlled by the position of the pycnocline (Hillaire-Marcel et al., 2004; Kozdon et al., 2009; Simstich et al., 2003; Xiao et al., 2014). In our data, we found DH always situated below the MLD, within the pycnocline. Thus, our observations confirm that a significant part of the calcification is likely to occur within the pycnocline, but the depth habitat of the species is not reflecting the depth of the local pycnocline.

Our observations indicate that *N. pachyderma* resides closer to the surface when sea ice and/or surface chlorophyll concentrations are high. The DH also increases with time since sea-ice break-up. This suggests that the DH of *N. pachyderma* is controlled by multiple, interacting variables, likely connected in the temporal dimension. The scheme in Fig. 9 summarizes our conceptual model: when either sea-ice cover or surface chlorophyll concentrations reach high values, *N. pachyderma* prefers shallower depths, while in open waters with low productivity levels, it lives deeper. While the relationship with sea-ice has been observed repeatedly (Carstens et al., 1997; Pados and Spielhagen, 2014), the relationship with surface chlorophyll at the surface is unexpected. Intuitively, rather than sea-ice and chlorophyll at the surface, the DH should reflect ambient conditions at depth. The DH does not appear to reflect the DCM (Fig. 9), but it could be that the species vertical abundance reflects the local depth at which a specific temperature or salinity optimum occur or where a given density is realised. We have thus extracted data on temperature, salinity and density at the depth of DH in all profiles where CTD data were available. The analysis reveals a large variability in all parameters, indicating that the DH is not tracking specific temperature, salinity or density (Fig.10). The observation that the subsurface depth habitat of *N. pachyderma* appears to be best predicted by surface parameters is counter-intuitive and points to an indirect relationship to the inferred surface drivers.

A possible link between surface properties and conditions at the DH could be light (or light-related processes). Increasing sea-ice cover and higher chlorophyll at the surface both act to reduce light penetration, potentially explaining why *N. pachyderma* habitat is shallow when either sea-ice or surface chlorophyll are high (Fig. 9). The exact mechanism by which the species would respond to light intensity is not clear. So far, there is no evidence that the species would possess

photosynthetically active symbionts. On the other hand, a recent molecular study indicated the presence of symbionts in a closely related species *Neogloboquadrina incompta* (Bird et al., 2018), and evidence for potential symbiosis with cyanobacteria in *Globigerina bulloides* (Bird et al., 2017) indicate that the range of symbioses in planktonic foraminifera may be more diverse than previously thought. However, half the observed DH values are > 100 m, indicating that a

substantial part of the population of the species inhabits depth where in the Arctic light for photosynthesis is not available (Ardyna et al., 2013). Alternatively, it could be that the vertical habitat of *N. pachyderma* reflects a compromising between living close to the DCM (finding food), but remaining in darkness (protected from predation). In many places of the ocean, heterotrophic protists are known to be metabolically more active at night (Hu et al., 2018), and predator evasion by remaining in darkness is the leading hypothesis explaining DVM in marine zooplankton (Hays, 2003). These hypotheses are

at present speculative and more investigations on the diet of *N. pachyderma* are needed for a better understanding of the process regulating its vertical distribution.

## 5 Conclusion

We compiled a dataset of 104 vertically resolved profiles of *N. pachyderma* concentration in the Arctic and North Atlantic

and analysed the relationship of the observed depth habitat to a range of potential biotic and abiotic drivers. The analysis confirms that *N. pachyderma* inhabits a wide portion of the water column, but its maximum concentration is typically found in the subsurface. The habitat depth is variable but most of the population is consistently found below the subsurface chlorophyll maximum. This indicates that the species is likely not grazing on fresh phytoplankton. The depth habitat of *N. pachyderma* as recorded by the vertically resolved plankton tow profiles shows no evidence for diel vertical migration or a

synchronised change in depth habitat with lunar cycle. Temperature, salinity and density alone (at the surface or at depth) do not show significant relationship with the habitat depth. Instead, sea-ice and chlorophyll concentration at the surface, in combination with the time since sea-ice break up explain almost a third of the variance in the depth habitat data. Most of the population of *N. pachyderma* resides between 50 and 100 m under dense sea-ice coverage and/or high surface chlorophyll concentration and the habitat deepens to 75 – 150 m when sea-ice cover is reduced and/or when chlorophyll in the surface is

low. This pattern reflects a response to an unknown primary driver acting below the DCM and likely reflecting trophic behaviour of the species, which is still poorly constrained. The knowledge gap on the ecological preferences of *N. pachyderma* is reflected in the mismatch the in the behaviour of *N. pachyderma* between observations and predictions by the PLAFOM2.0 model. Our findings can serve as a basis to calibrate new ecosystem models and refine paleoclimatic reconstructions based on *N. pachyderma* in the Arctic and its adjacent seas. Our analysis rejects the hypothesis that the

vertical habitat of the species is tied to the DCM and the existence of a significant relationship with sea ice and surface chlorophyll allows us to derive a model that can predict the habitat depth of the species across the Arctic realm.



*Author contributions*. MK, LJ, and MG designed the study. KK provided the PLAFOM2.0 data. MG generated the data and carried out the analyses. All authors contributed to writing the paper.

*Competing interests*. The authors declare that they have no conflict of interest.

*Acknowledgements*. The master and crew of the F.S Maria S. Merian are gratefully acknowledge for support of the work during the MSM09/2 cruise. This project was supported by the Deutsche Forschungsgemeinschaft (DFG) through the International Research Training Group "Processes and impacts of climate change in the North Atlantic Ocean and the Canadian Arctic" (IRTG 1904 ArcTrain).

*Data availability*: Total concentrations and filtered volumes will be made available on request to the main author until their online publication on PANGAEA (https://pangaea.de/). The table complete with data source and derived environmental data of the stations included in the study is available on Zenodo (DOI:10.5281/zenodo.2585796)

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



**Table 1** Results of the *t*-test to performed on the samples collected in normal day/night conditions to assess the effects of DVM on DH.

| Time of the day | n | Mean DH (m) | Std. Deviation | t-value | p-value |
|---|---|---|---|---|---|
| *Night* | 19 | 99.069 | 46.762 | -1.82 | 0.08 |
| *Day* | 9 | 66.949 | 35.401 | | |





**Table 2** Results of the periodic regression performed to asses the influence of the lunar cycle on DH.

| Predictors | Depth habitat (m) | |
| --- | --- | --- |
| | *Estimates* | *p* |
| sin (Lunar day $^R$) | -8.41 | 0.171 |
| cos (Lunar day $^R$) | -10.39 | 0.071 |
| *Observations* | 104 | |
| $R^2$ / adjusted $R^2$ | 0.047 / 0.029 | |





**Table 3** Results of the multiple regression model including sea-ice concentration, chlorophyll concentration at surface and time since sea-ice break-up as predictors.

| Predictors | DH (m) | | | log$_{10}$(DH) (m) | | |
| --- | --- | --- | --- | --- | --- | --- |
| | Estimates | CI | p | Estimates | CI | p |
| (Intercept) | 110.76 | 80.37 – 141.15 | **<0.001** | 2.03 | 1.89 – 2.18 | **<0.001** |
| Sea-ice (%) | -0.04 | -0.08 – -0.00 | **0.033** | 0 | -0.00 – -0.00 | **0.021** |
| Chlorophyll at surface (mg m$^{-3}$) | 10.94 | -10.82 – 32.71 | 0.329 | 0.06 | -0.04 – 0.16 | 0.263 |
| Days after sea-ice break-up | 0.71 | 0.22 – 1.20 | **0.007** | 0 | 0.00 – 0.01 | **0.005** |
| Interaction (Chlorophyll and sea-ice break-up timing) | -0.81 | -1.25 – -0.37 | **0.001** | 0 | -0.01 – -0.00 | **<0.001** |
| Observations | 52 | | | 52 | | |
| R$^2$ / adjusted R$^2$ | 0.343 / 0.287 | | | 0.388 / 0.336 | | |



**Figure captions**

**Figure 1.** Plankton net stations with vertically resolved *N. pachyderma* counts that were used in this study. Background colour indicates the mean summer sea surface temperature (SST) (data from World Ocean Atlas 2013).

**Figure 2.** Temporal and environmental coverage of the vertical profiles of *N. pachyderma* concentration included in the study. The distribution of (a) the months and (b) days of the synodic lunar cycle of sample collection, showing a summer bias but even coverage of the lunar cycle. The relationship between the environmental conditions during sample collection (c-d) indicate the extent of the sampled environmental space.

**Figure 3.** Example of vertical profiles from three stations included in the study displaying shallow (left), intermediate (centre) and deep (right) depth habitat (DH).

**Figure 4.** a) Relationship between the depth habitat (DH) and the average living depth (ALD). The dashed red line shows the
linear fit while the solid line represent the 1:1 relationship between the two variables. b) Relationship between the DH and the PC1 resulted from the PCA calculated on the normalized counts. The normalized density profiles in the plot show the relationship between the loadings on the PC1 and the shape of the distribution. The dashed red line shows the linear fit.

**Figure 5.** a) Correlation between depth habitat (DH) and the environmental variables calculated in all the sites, in the subset
with sea-ice and without sea-ice (only sites where all the tested variables were available were considered). Chl= Chlorophyll concentration at surface, Sea_ice= Sea-ice coverage, DCM= Depth of Chlorophyll maximum, SST= sea surface temperature, MLD = depth of the mixed layer and SSS= sea surface salinity. b) Relationship between DH and sea-ice concentration in the stations covered by sea-ice (all the sites with available sea-ice data are shown, n=65). c) Relationship between DH and chlorophyll concentration at the surface for the sea-ice free stations (all the sites with available chlorophyll data are shown,
n=22).The dashed red lines show the linear fit.

**Figure 6.** Relationship between depth habitat (DH) and the time (days) after the sea-ice break-up. The dashed red line shows the linear fit.

**Figure 7.** Comparison of observed DH and the PLAFOM2.0 predictions.

**Figure 8.** a) Relationship between the DH predicted by PLAFOM2.0 and a) sea-ice concentration in the stations covered by sea-ice and b) between DH predicted by PLAFOM2 and chlorophyll concentration at the surface for the sea-ice free stations (values averaged for the months June, July, August and September). The dashed red lines show the linear fit.



**Figure 9.** Data-based scheme of the final model: samples are displayed in descending order for sea-ice concentration (light-blue fading bar) and ascending chlorophyll concentration (green fading triangle) to simulate the time dimension. The green star symbols (*) represent the depth of the chlorophyll maximum and the dashed red line show the smooth fit of the data.

5   **Figure 10.** Conditions of a) temperature, b) salinity and c) density at the DH (top) and in the first 600 meters of water column (bottom) for all the sites with available CTD data.





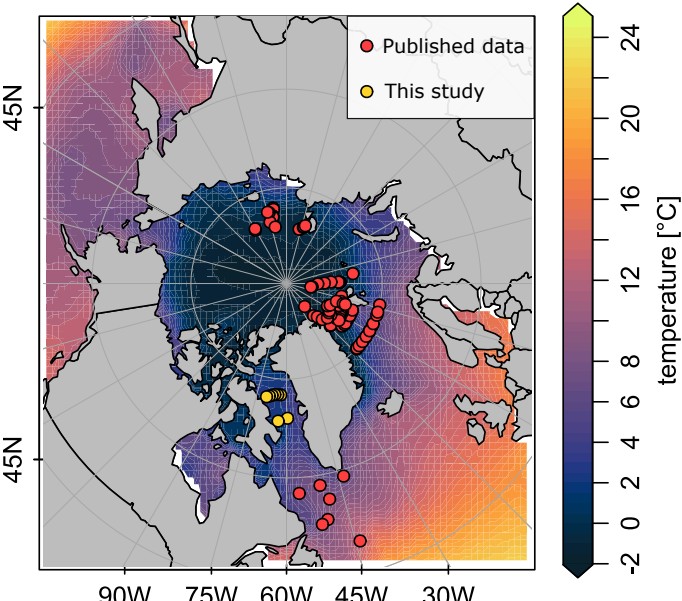





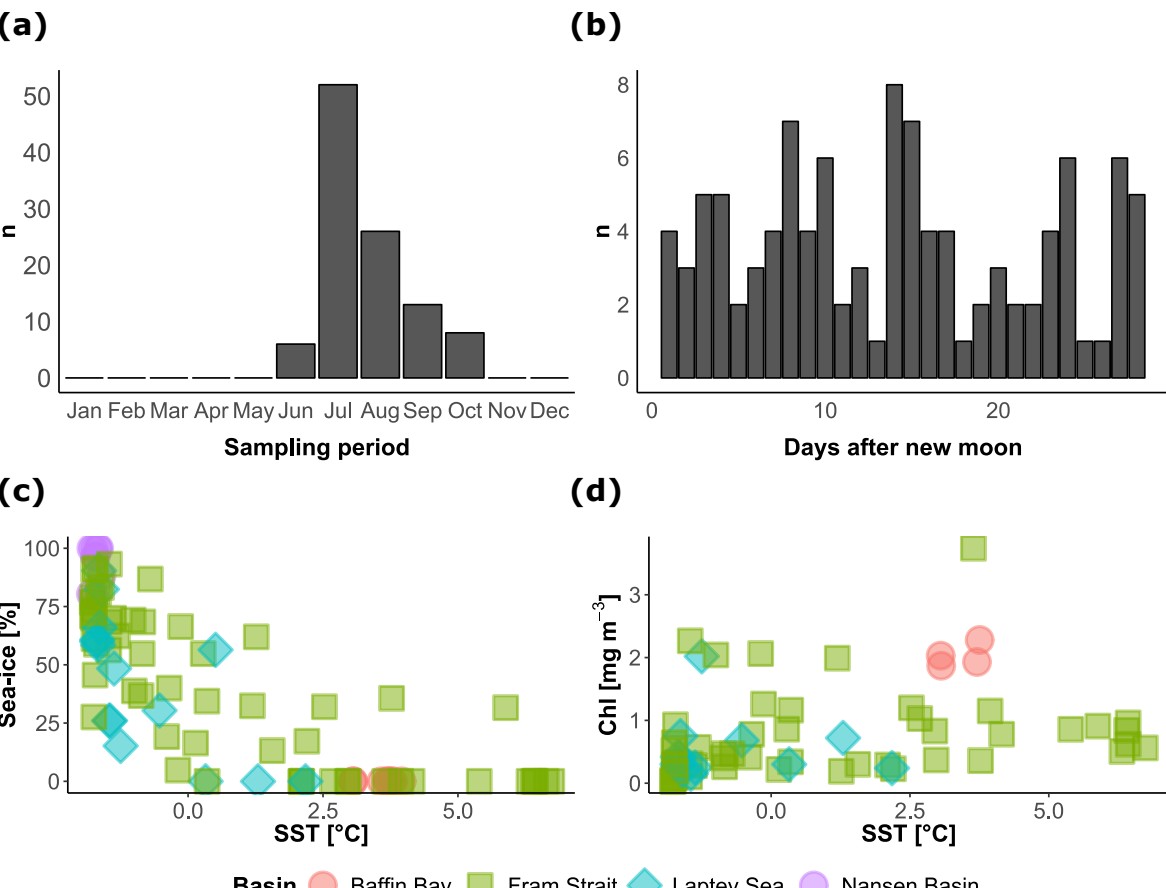





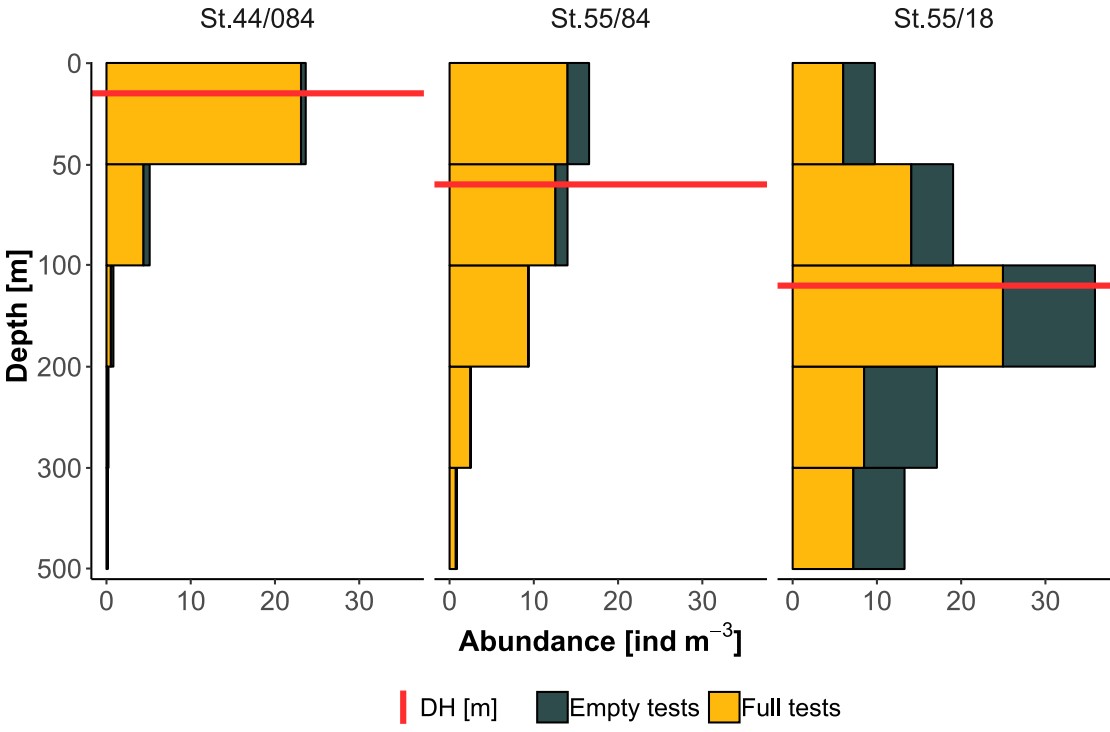





**(a)**                          **(b)**

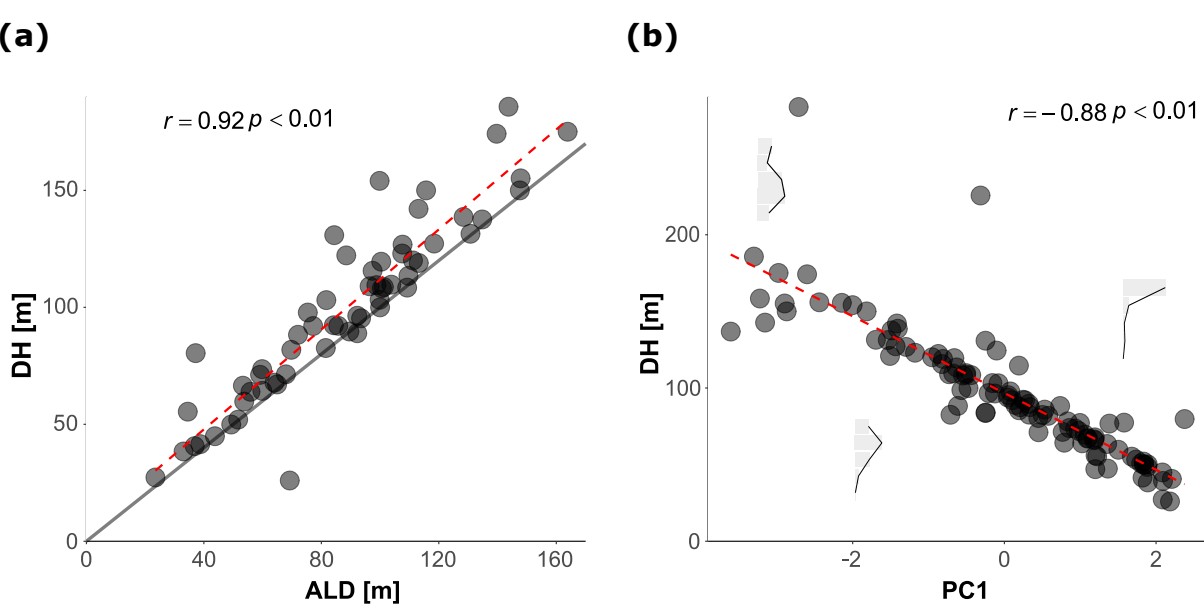





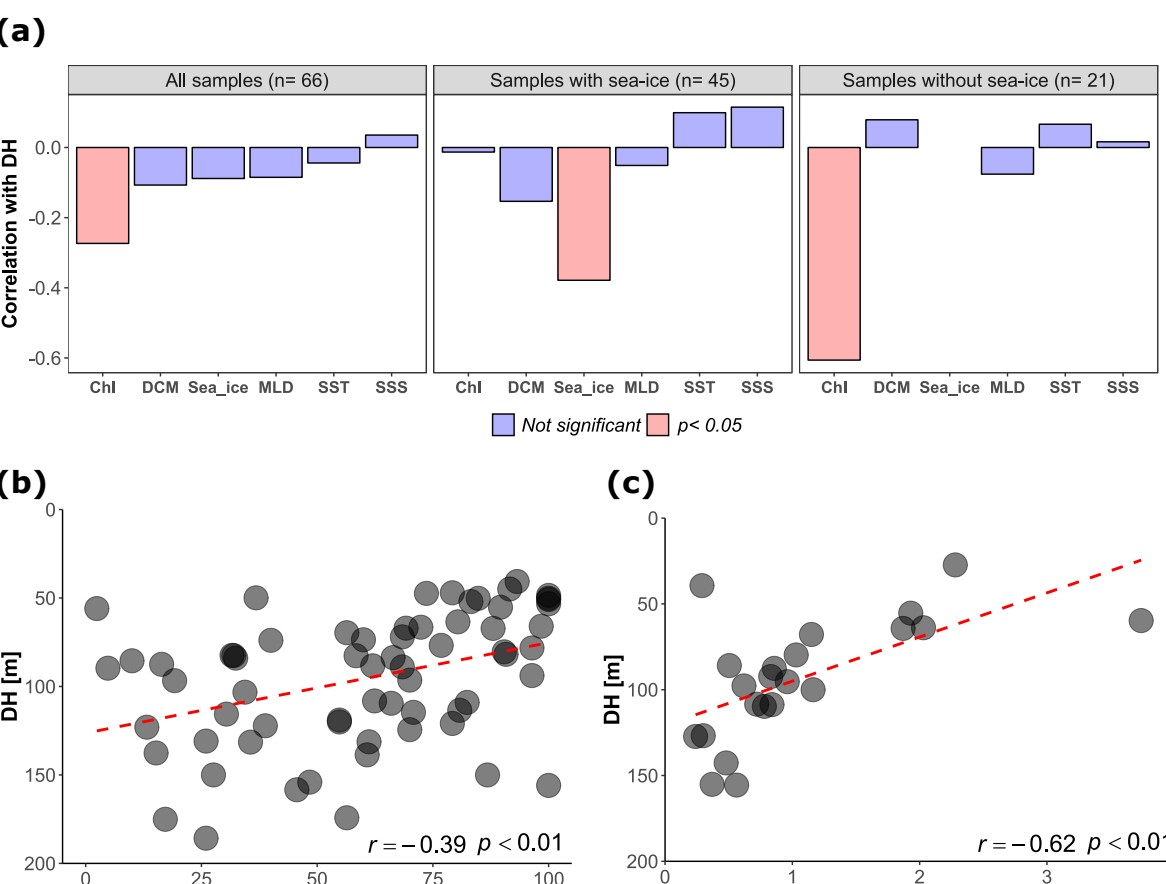





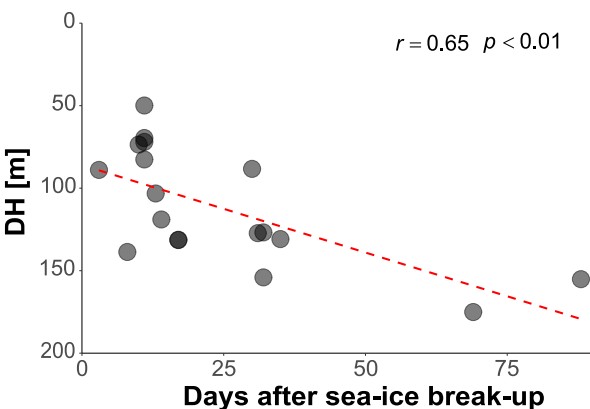





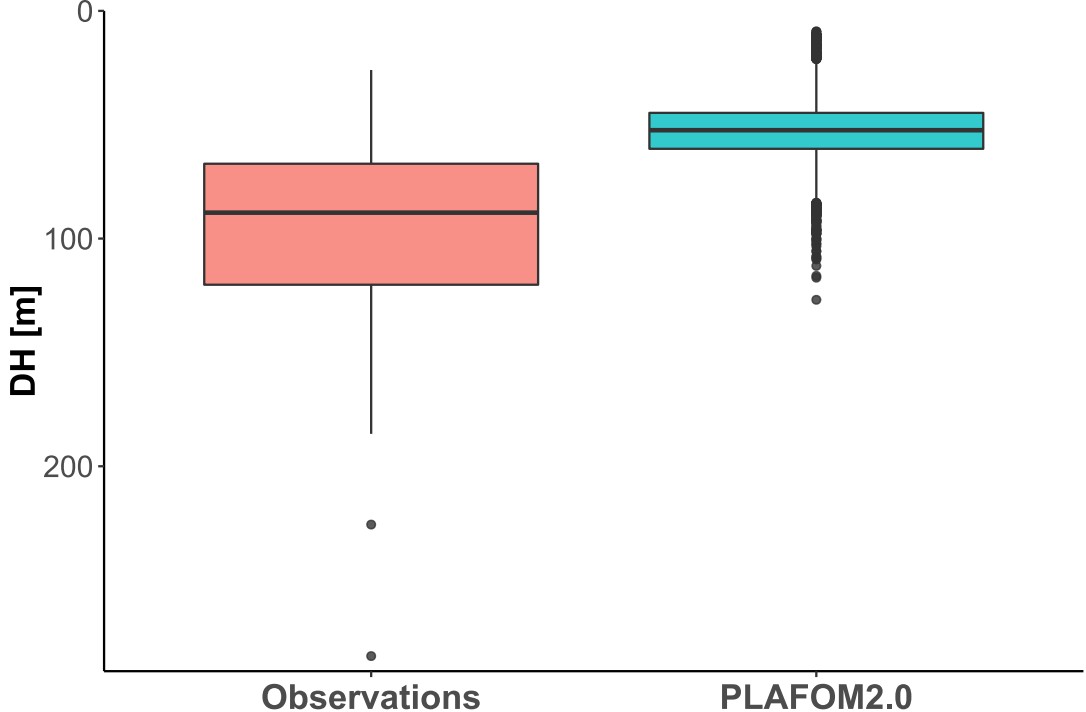





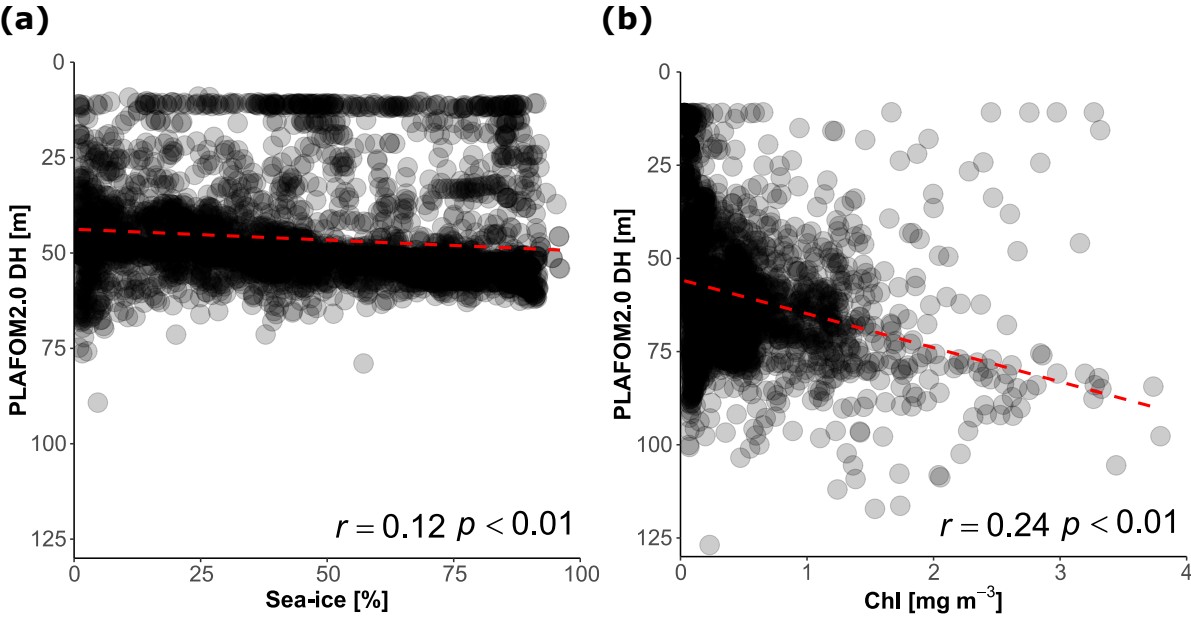





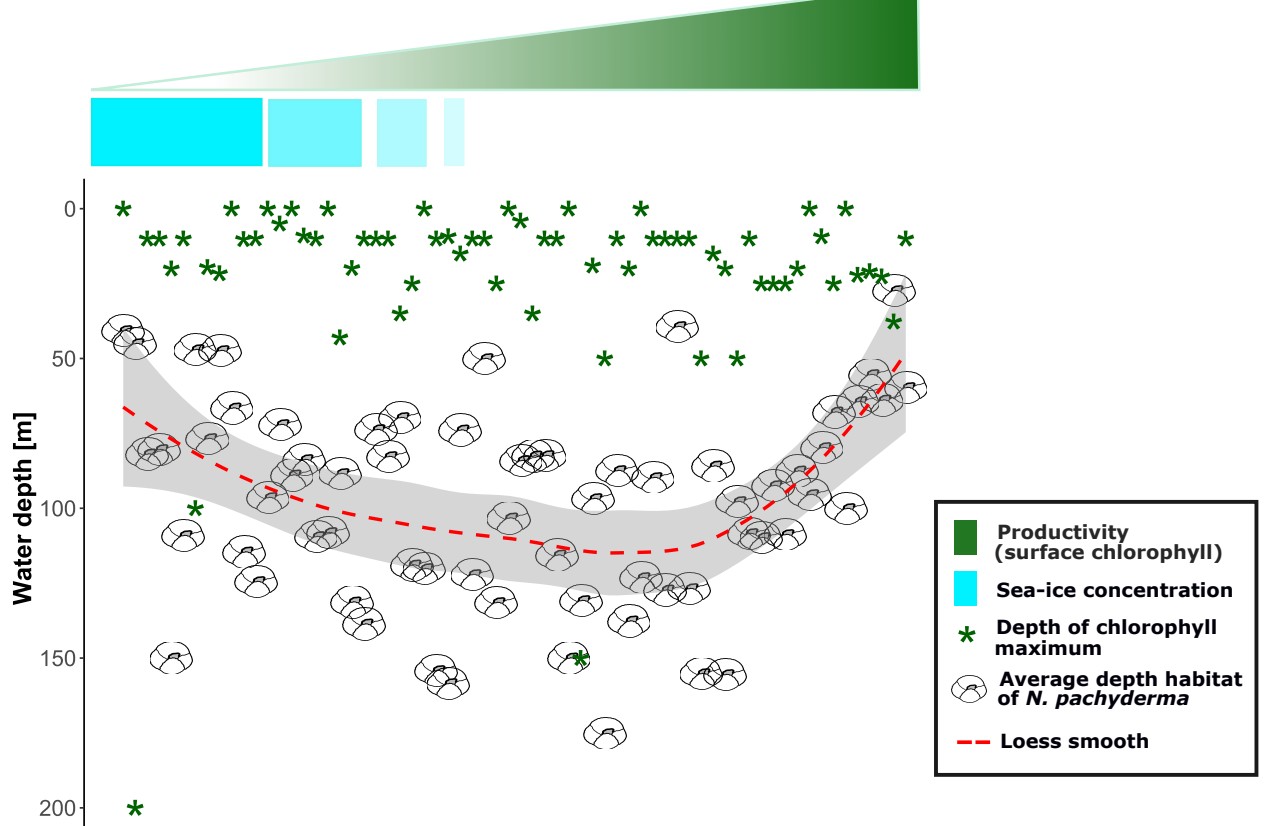





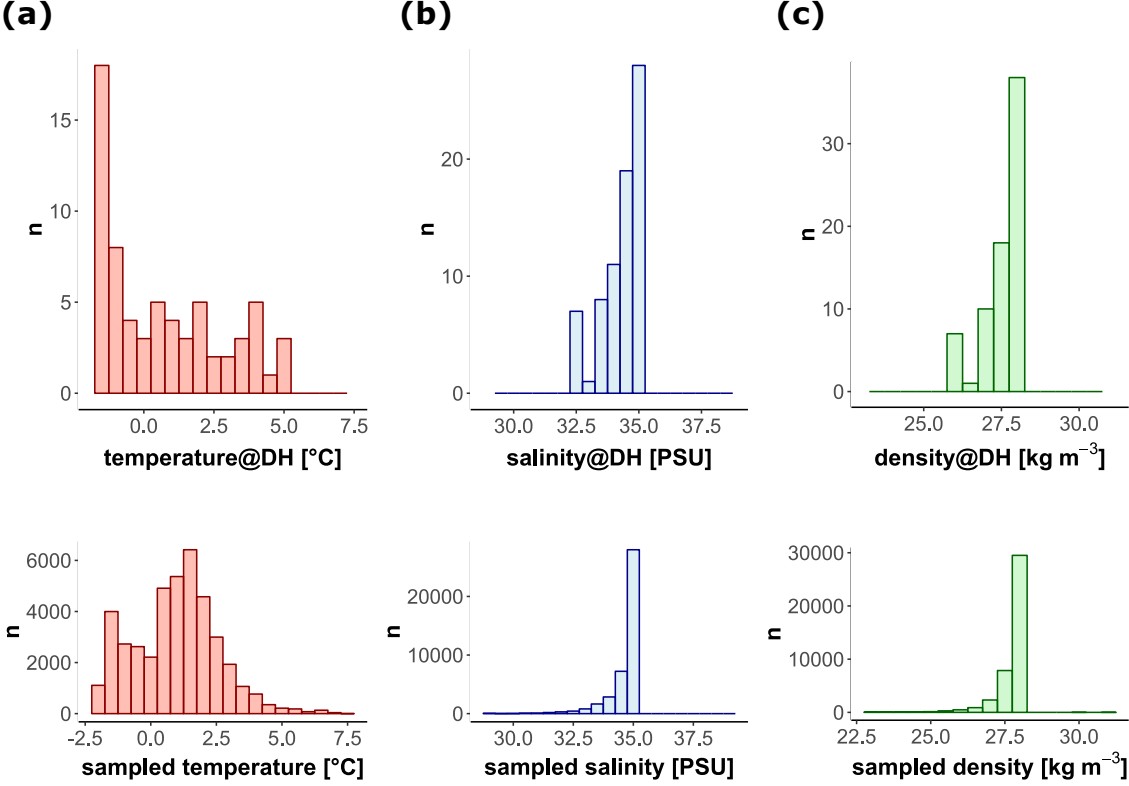