# Peer review of "Depth habitat of the planktonic foraminifera *Neogloboquadrina pachyderma* in the northern high latitudes explained by sea-ice and chlorophyll concentration"

_Biogeosciences, 2019_

## Referee Comment (RC1) · Anonymous Referee #1 · 26 Mar 2019

Greco et al. present an interesting study on the variability of depth habitat of the planktonic foraminifera N. pachyderma, the most important species in the Arctic. Due to the ubiquity of N. pachyderma both in paleo-records and in present-day Arctic and the significance of its depth habitat for paleoreconstructions, the authors address a relevant scientific question within the scope of BG. The presented results can be used in paleoreconstructions as long as there are proxy on chlorophyll and sea-ice concentration available. The authors compile new and existing data from the Arctic and the North Atlantic Ocean and the substantial conclusions that they come up with are

also novel. The scientific methods and assumptions are valid and clearly outlined and the results are sufficient to support the interpretations and conclusions. The authors compare the observational data with a numerical model though this comparison only shows that the model does not perform very well. The methods are described sufficiently precisely. However, as I am not an expert on statistics, I cannot evaluate this aspect of the manuscript. The authors give proper credit to related work and clearly indicate their own contributions. The title clearly reflects the contents of the paper and the abstract provides a concise and complete summary. The MS is well-structured and written and the language is fluent and precise. Therefore I find the MS suitable for publication in Biogeosciences after minor revisions according to general, specific and technical comments listed below. I am looking forward for the authors' response and further discussion.

General comments

The authors use the term 'habitat depth' along with 'depth habitat (DH)' which is a bit confusing. Are these two different terms? If so, what is the difference between them? Wouldn't it be better to stick to only one of these terms? I don't see a significant difference between them.

A table listing all the published profiles used in the study and/or a more detailed location map would be useful, at least as an appendix or supplementary material. Now it is completely unclear what published data are you using.

A weak, though unavoidable, point of the study is that it compiles data with different sampling depth intervals which might bias the calculated DHs. The authors should stress and discuss this issue a bit more.

Specific comments

2.3 (page 2, line 3) and 2.20: I know that 'climate change' is a catchy phrase but N. pachyderma is a marine species and so it doesn't directly react to climate changes

but rather to changes in marine environment (which, of course, are usually related to climate changes). Please be more precise in your wording!

2.22: Please change 'Arctic and its marginal seas' to either 'Arctic Ocean and its marginal seas' or just 'Arctic' (or 'Arctic seas').

4.3: similar as above

4.30-32: It is not clear whether the satellite data were used only for data generated by the authors or also for the data from the literature. Please explain.

7.8: In the text the adjusted r2 = 0.32, while in Table 3 it's 0.336 ≈ 0.34. Please correct or explain the difference.

8.27: It might not be clear to a reader whether 'lowering the DCM' means lowering the value of the DCM, i.e. moving it up the water column (shallowing) or lowering it 'geometrically', i.e. moving it down the water column (deepening, which I guess is the case). Please clarify.

9.26: 'at the depth of DH' please rephrase.

24: The small diagrams in Fig. 4b (normalized density profiles?) need more explanation.

Technical comments

2.32: I'm not sure about the rules concerning citing of papers with three authors in Biogeosciences but shouldn't it be just 'Ding et al., 2014'?

5.9: Table 2 is referenced in the text before Table 1. Again I am not sure about the rules in BG, but I guess you should change the numeration.

5.14: I suppose 'Fig. 2d' was meant.

5.33 & 6.1-2: You already introduce the DVM abbreviation so use it!

6.14: Use 'DH' instead of 'depth habitat'

9.2: 'sea-surface' instead of 'seas-surface'

10.27: An unnecessary 'the' after 'mismatch'.

---

## Referee Comment (RC2) · Robert F. Spielhagen (Referee) · 28 Mar 2019

General comments

Planktic foraminifera of the species Neogloboquadrina pachyderma are a major carrier of paleoenvironmental information in Arctic and sub-Arctic marine sediments and widely used to reconstruct properties of the upper water column, namely sea-ice coverage, salinity, and temperature. The variability of their geochemical composition (e.g., stable isotopes, Mg/Ca) can be very large within an investigated sediment core. In

particular to interpret this variability it is important to know which factors may determine the habitat depth of the species. A number of studies have determined the depth distribution of N. pachyderma under a large variety of boundary conditions (oceanic parameters). A synoptic study involving all the available depth distribution data plus oceanic data from the same sites was much needed but still lacking. The manuscript by Greco et al. fills this gap with a statistical evaluation of mostly published data from the Arctic Ocean and its neighboring seas, amended by some new data from Baffin Bay and results from a numerical model. The combination of biological, physical oceanographic, biochemistry and modelling data results in a novel approach to determine the habitat depth of N. pachyderma on a larger scale and is thematically well suited for the journal Biogeosciences. It is well-written in very good English. The structure of the manuscript follows standard principles of scientific publications. The abstract gives a good overview of the topic, the methodological approach, the major results and the main conclusions. The Introduction chapter gives a good overview of the present knowledge and thereby manifests the problem of defining the factor(s) determining the depth habitat of N. pachyderma. It also describes the general approach applied here and which environmental factors are considered as potentially responsible for the variability in habitat depth. The Material and Methods chapter describes in detail the origin of the data sets used for the following evaluation, the methods to obtain the new data from Baffin Bay, and the statistical methods applied to evaluate and weight the environmental factors determining the habitat depth. I am not an expert on such statistical methods and can therefore not evaluate whether proper attention has been paid to significance levels. The Results chapter lists briefly but in sufficient detail the major outcome of the statistical evaluations, in particular the correlation of habitat depth to individual and combined environmental parameters and how the results from statistical evaluations compare to the model results. The Discussion chapter puts the results of statistical evaluations in context and elaborates which environmental facors are determining the habitat depth. The outcome is discussed with respect to previous hypotheses on which parameters have forced N. pachyderma to live shallower or

deeper in the water column. Interestingly, some of these published hypotheses (which in most cases were based on regional studies) are not supported by the conclusions of Greco et al.. The very large data base of the present study (I cannot see that relevant published data sets were left out) is the advantage of the present study and adds significantly to the credibility of the conclusions presented here. To me, the discussion appears to the point and overall sound, and I cannot see that systematic errors may bias the conclusions. These are compiled in the Conclusion chapter which lists the major findings but also open questions which may trigger further research in this field. The figures and tables are mostly clear and easy to understand (see comments below for minor exceptions). Overall, I think this paper is already in a mature state and does not need significant changes. Publication in Biogeosciences is recommended after some minor revision in response to the points listed below.

Specific comments to the authors

Title and manuscript text: Regarding the use of "planktonic" (instead of "planktic") in this manuscript I suggest to read the advice of the Godfather of Paleoceanography, Cesare Emiliani, which can be found here: https://www.cambridge.org/core/journals/journal-of-paleontology/article/plankticplanktonic-nekticnektonic-benthicbenthonic/CDF06242F0F9130B7A5A082DFDDFC425

You use both "habitat depth" and "depth habitat" in the manuscript. The latter is defined on page 5 (line 17ff), the first not. Do these terms have different meanings? If yes, you should explain this. I note that even at the end of the manuscript, in the Conclusion chapter, you still use both terms (page 10, lines 17/18). That is confusing!

It will be helpful for the reader to receive a bit more information on the PLAFOM2.0 model. As it stands, we just learn that it can predict the seasonal and vertical habitat of N. pachyderma. For those readers who have not studied the Kretschmer et al. (2018) paper in detail, you should use 2-3 lines to explain what the model is based on and

which boundary conditions are used.

Many readers may not be acquainted with all the statistical parameters applied to determine correlations, anticorrelations, significance limits etc.. Those terms used widely throughout the manuscript (e.g., r, p, R, F-test, t-test) should be explained in the manuscript, including a comment on what higher or lower values mean.

I suggest not to mix British and American spelling. Either use "paleo" and "...ize" or "palaeo" and "...ise". Please check the entire manuscript for other language cases (e.g., "metres" vs. "meters").

Comments by page and line numbers (page/line)

2/1: Better write "dominant plankt(on)ic foraminifer species" 2/8: Arctic and North Atlantic oceans 2/10: Here and in other places you mention "chlorophyll a", later you also just write "chlorophyll". Be precise in what you mean. 2/23-24: "When the organism dies, its calcite shells sink to the seafloor and when preserved in the sediments, they serve ..." Do not mix singular and plural.

3/10-14: You are discussing the issue of diel vertical migration again on page 7, lines 18ff, largely repeating what is said here. I suggest to shorten this part in the introduction and put the discussion where it belongs. 3/22: drivers 3/22-26: Very long sentence, hard to read. Split it into two.

4/17: with a conductivity 4/18: obtain vertical profiles 4/19: for all stations 4/20: chlorophyll a concentration profiles 4/21: from the PANGAEA

5/1: all stations 5/2: time of collection 5/11: related to SST

6/23: neither in the complete data

8/21: relationship between DH and environmental parameters 8/21-23: Three sentences starting with "This...". Maybe rephrase? 8/25ff: Better write "In the model, this overestimation of the MLD affects..." 8/34: matter

9/1: depth of 9/2 sea surface 9/6: tolerance limit 9/5-7: Split long sentence into two.

10/2-3: ...evidence ... indicates 10/6-7: compromise between ... and ... 10/22-25: Split long sentence into two. 10/27: mismatch in the

11/6: gratefully acknowledged

12/3-4: Delete blanks! 12/10: ocean 12/13: Carstens, J. 12/13: Sarnthein

13/4: Delete "(Ehrenberg 1861)"

16: A word is missing in the table caption!

Fig. 2c/d: Several symbols are hidden. Possibly use open symbols with no filling?

Fig. 9: Why is "Productivity" related to a filled symbol in the legend while the triangle is open in the figure?

---

## Referee Comment (RC3) · Antje Voelker (Referee) · 29 Mar 2019

Greco and co-authors compiled new and published vertical abundance data from multi-net tows to evaluate – using statistical approaches – the habitat depth of polar foraminifera N. pachyderma and its relationship to environmental parameters. The study provides new and important insights into a species widely used in paleoceanographic reconstructions, but still with limited information on its living conditions. The authors compare their evidence also to the outcome of the PLAFOM2.0 model (with limited success). With the environmental changes currently occurring in the subpolar

[Figure]

North Atlantic and Arctic Ocean this study is for sure timely and relevant for any future studies. The manuscript is well written, the data well presented and deserves to be published in Biogeosciences after minor revision.

The following are more general comments that might help improve the manuscript, but are not essential for accepting the manuscript:

1) There exists a very nice study (PhD thesis) [in German] on "The planktonic foraminifera Neogloboquadrina pachyderma (Ehrenberg) in the Weddell Sea, Antarctica" by Doris Berberich published as Berichte zur Polarforschung 195, in 1996. Although this is a different genotype than in the northern hemisphere, it seems that some aspects of the Greco et al. and Berberich observations are similar. So I urge the authors to have a look at this work. I do not know, if the authors could verify with their data is the deeper depth habitat in their data is also related to more adult/ terminal stage specimens and thus potentially to the reproduction cycle. Berberich is also discussing influence of phytoplankton abundance (i.e., food supply) on the foraminifera abundance and sees similar changes in depth as discussed on p. 9 lines 17 to 30. She is referring to Arikawa (1983) when discussing the relationship between N. pachyderma abundance and the deep chlorophyll a maximum. So the Arikawa study is another one the current authors should look into as support for their observation that the depth habitat of their genotype of N. pachyderma appears to be below the chlorophyll maximum. Arikawa, R. (1983), Distribution and taxonomy of Globigerina pachyderma (Ehrenberg) off the Sanriku Coast, Northeast Honshu, Japan. Tohoku Euniv. Sci. Repts., Ser. 2 (Geol.), 53, p. 103-157

2) p. 4 line 29: did the authors inquire at the AWI oceanography group if the CTD data collected during the ARK campaigns might have been stored there? Since I participated in ARK-X/2, I verified the cruise report and it clearly says on page 95 that at most stations with plankton sampling hydrographic information was obtained with a CTD probe.
More detailed comments to the manuscript itself:

1) throughout the manuscript you are referring to the North Atlantic, even though your samples are actually limited to the subpolar and polar regions of the North Atlantic. If you do not want to use the term Nordic Seas (for the area between Iceland, Greenland, Norway and Svalbard), you could use "northern North Atlantic" to better describe the geographical range of your samples.

2) p. 3 line 26: why is food source/supply not mentioned here -although one could argue that this could be a consequence of the change in the environmental conditions?

3) Material: please provide a table with the stations, date/ year of collection, data source for published data. From your figures one can deduce the season etc., but not how the samples are distributed over the years. Please also provide the name of the station excluded from the Jensen (1998) data set.

4) p. 4 line 18: please provide the depth until which pigment concentrations were measured. Were the profiles also done down to 300 m?

5) p. 5 line 11: small English correction; it should say "related to"

6) p. 7 line 15: it would be good if you could provide the reader with the information how and in which geographical resolution sea ice and chlorophyll are presented in the earth system model, from which PLAFOM2.0 derives its environmental conditions. I wonder if the poor relationship between observations and model might be a resolution problem or sea ice itself not being presented in the model.

7) p. 9 line 14: if the authors would like to include a study more concentrated on isotopic evidence from the Arctic Ocean they could add the following reference: could also look into Hillaire-Marcel, C., 2011. Foraminifera isotopic records. . . with special attention to high northern latitudes and the impact of sea-ice distillation processes. IOP Conference Series: Earth and Environmental Science 14, doi:10.1088/1755-1315/14/1/012009

8) p. 9 line 30: although the authors write on p 10 line 18 that the species is likely not grazing on fresh phytoplankton, I wonder if type of food source might not be a driver with a preference for "fresh food" during period with a shallower DH and more refracted organic matter during periods when the species prefers the depths below the chlorophyll maximum.

---

## Referee Comment (RC4) · Caterina Bergami (Referee) · 7 Apr 2019

The paper provides interesting paired planktonic foraminfera and environmental data from an important oceanographic region both from new and yet published data in order to better understand the habitat depth preferences of N. pachyderma. This type of studies can help to better understand in which way environmental parameters control the habitat depth and behaviour of this important planktonic calcifier and has an impact on future palaeoecological and palaeoceanographic studies in this area and in other high latitude environments. The authors also compare their evidence to the outcome of

the PLAFOM2.0 model, with limited results. The manuscript is well structured and the data well presented and relevant for future studies on the same issues. The amount of figures and tables is adequate to illustrate the results discussed in this paper. The paper deserves to be published in Biogeoscience after some minor revisions that I listed below and in the attached pdf file. I also suggest to the author a further check of the English language.

Technical points:

- please check the use of the acronyms along the text. Once you define them the first time, please use the same along the text (check in particular DVM, SST and SSS).

- choose between the term "habitat depth" and "depth habitat (DH)" as definied in the text and use it accordingly. - please check along the text the use of the term "compilation". I would prefer dataset.

- Material and methods sections (from page 5/line 16 to page 6/line 11: In this part of the text some results are mixed with M&M. Please, check and move the results to the following section.

- Page 4/line 3-4: "We retained all other profiles, despite the differences in the mesh size, counted size fraction and vertical resolution". This phrase is not clear, please re-phrase. What do you mean?

- page 9/line 11-12: "sedimentary and plankton specimens". Do you mean fossils and living specimens?

Please also note the supplement to this comment:
https://www.biogeosciences-discuss.net/bg-2019-79/bg-2019-79-RC4-supplement.pdf

[Figure]

**Supplement:**

[revised manuscript text omitted]

---

## Referee Comment (RC5) · Katrine Husum (Referee) · 8 Apr 2019

General comments

The manuscript BG_2019_79 by Greco et al. study which factors that influence the depth habitat of the planktic foraminifera Neogloboquadrina pachyderma using both published and new data together with a suite of statistical methods. The scope of the study is very timely as the regions where this species dominates are subject to climate-ocean changes, hence in order to evaluate current changes and establish robust natural baseline values a better understanding of this species' depth habitat is necessary. The manuscript is well-written and in an advanced state.

Specific comments

1. Figure 10: Introduce this information and figure early on?

2. It would be beneficial to define what is a good correlation/a correlation/a weak correlation, e.g. what is the difference between the r- values of -0.28; -0.38 and -0.60. I am not an expert on statistics but +/- 0.28 seems like a weak correlation? It would also be good to point out that it seems that the correlation improves with a smaller number of observations (e.g. 21 samples without sea-ice)?

3. The PLAFOM2.0 model is introduced somewhat superficially; more information would be useful.

Technical comments

1. Overall: The use of "planktonic" vs "planktic". Please refer to Emiliani 1991: https://doi.org/10.1016/0377-8398(91)90003-O

2. Overall: There are many acronyms in the paper. Except for SST, SSS, DVM and DH they do not help reading the paper.

3. Page 5, lines 6-15 and lines 20-32 ("Materials and methods"): This seem more like a description of results, which it may benefit to move to the start of "Results".

---

## Author Comment (AC1) · 9 May 2019

**Response to Reviewer #1**

*We would like to take this opportunity to thank the anonymous reviewer for their helpful comments on our manuscript. Below we provide a detailed response to their comments (in italics), indicating the changes that have been made. Line numbers refer to those of the revised manuscript that includes all tracked changes.*

*With kind regards,*

*Mattia Greco (on the behalf of all co-authors)*

Greco et al. present an interesting study on the variability of depth habitat of the planktonic foraminifera N. pachyderma, the most important species in the Arctic. Due to the ubiquity of N. pachyderma both in paleo-records and in present-day Arctic and the significance of its depth habitat for paleoreconstructions, the authors address a relevant scientific question within the scope of BG. The presented results can be used in paleoreconstructions as long as there are proxy on chlorophyll and sea-ice concentration available. The authors compile new and existing data from the Arctic and the North Atlantic Ocean and the substantial conclusions that they come up with are also novel. The scientific methods and assumptions are valid and clearly outlined and the results are sufficient to support the interpretations and conclusions. The authors compare the observational data with a numerical model though this comparison only shows that the model does not perform very well. The methods are described sufficiently precisely. However, as I am not an expert on statistics, I cannot evaluate this aspect of the manuscript. The authors give proper credit to related work and clearly indicate their own contributions. The title clearly reflects the contents of the paper and the abstract provides a concise and complete summary. The MS is well-structured and written and the language is fluent and precise. Therefore I find the MS suitable for publication in Biogeosciences after minor revisions according to general, specific and technical comments listed below. I am looking forward for the authors' response and further discussion.

General Comments

The authors use the term 'habitat depth' along with 'depth habitat (DH)', which is a bit confusing. Are these two different terms? If so, what is the difference between them? Wouldn't it be better to stick to only one of these terms? I don't see a significant difference between them.

*The reviewer rightly mentions that this is confusing, the two terms refer to the same parameter; we will correct and homogenize the terminology adopting only the term "depth habitat".*

A table listing all the published profiles used in the study and/or a more detailed location map would be useful, at least as an appendix or supplementary material. Now it is completely unclear what published data are you using.

*We appreciate this comment. Due to the size of such a table we prefer to make it available as supplementary material at zenodo.org, where long-term storage is guaranteed. We would like to point out that the link to the table with the complete metadata and environmental data was already provided in the "Data availability" section in the original manuscript.*

A weak, though unavoidable, point of the study is that it compiles data with different sampling depth intervals which might bias the calculated DHs. The authors should stress and discuss this issue a bit more.

*The reviewer rightly points out that the precision with which the DH can be determined depends on the vertical resolution of the individual casts. By mixing casts with different vertical resolution we unavoidably lose some precision, but we would argue that this introduces random noise, rather than a*

*systematic bias. This is probably part of the reason why our predictive models do not explain all the variability in DH. We will add some discussion on this in the method section at page 6, lines 7-9:*

*'Anyway, since the accuracy with which the DH is determined is linked to the vertical resolution of the single profiles, mixing casts with different vertical resolution causes unavoidably the loss of some precision and the introduction of random noise in the data.'*

Specific comments

2.3 (page 2, line 3) and 2.20: I know that 'climate change' is a catchy phrase but N. pachyderma is a marine species and so it doesn't directly react to climate changes but rather to changes in marine environment (which, of course, are usually related to climate changes). Please be more precise in your wording!

*We will change the sentence to 'To assess the reaction of this species to a future shaped by climate change and to be able to interpret the paleoecological signal contained in its shells…'*

2.22: Please change 'Arctic and its marginal seas' to either 'Arctic Ocean and its marginal seas' or just 'Arctic' (or 'Arctic seas').

*We will change 'Arctic and its marginal seas' to Arctic Ocean and its marginal seas.*

4.3: similar as above

*We will change Eurasian Arctic and its marginal seas to Eurasian Arctic Ocean and its marginal seas.*

4.30-32: It is not clear whether the satellite data were used only for data generated by the authors or also for the data from the literature. Please explain.

*We rewrote the sentence to avoid confusion 'In addition to the in-situ data, daily sea ice concentrations for each location of all the 104 sites included were extracted from 25 × 25 km resolution passive microwave satellite raster imagery obtained from the National Snow and Ice Data Centre'*

7.8: In the text the adjusted r2 = 0.32, while in Table 3 it's 0.336 - 0.34. Please correct or explain the difference.

*We apologise for this mistake, the number referred to in the table is correct and we have changed the text accordingly.*

8.27: It might not be clear to a reader whether 'lowering the DCM' means lowering the value of the DCM, i.e. moving it up the water column (shallowing) or lowering it 'geometrically', i.e. moving it down the water column (deepening, which I guess is the case). Please clarify.

*We modified the sentence replacing 'lowering' with 'deepening'.*

9.26: 'at the depth of DH' please rephrase.

*We rewrote the sentence using 'at the level of DH'.*

24: The small diagrams in Fig. 4b (normalized density profiles?) need more explanation.

*These profiles only serve to illustrate the meaning of loadings of the first principle component. To make this clearer we rewrote the sentence in the caption: 'The density profiles based on the standardized counts in the plot show examples of shape of the vertical distribution of N. pachyderma at three PC1 loadings.'*

Technical comments

2.32: I'm not sure about the rules concerning citing of papers with three authors in Biogeosciences but shouldn't it be just 'Ding et al., 2014'?

*The reviewer is correct and we have changed it accordingly.*

5.9: Table 2 is referenced in the text before Table 1. Again I am not sure about the rules in BG, but I guess you should change the numeration.

*We will correct the reference to Table 1 as we refer to the DVM results.*

5.14: I suppose 'Fig. 2d' was meant.

*Correct and amended.*

5.33 & 6.1-2: You already introduce the DVM abbreviation so use it!

*Done.*

6.14: Use 'DH' instead of 'depth habitat'

*Done.*

9.2: 'sea-surface' instead of 'seas-surface'

*Done.*

10.27: An unnecessary 'the' after 'mismatch'.

*Done.*

---

## Author Comment (AC2) · 9 May 2019

**Response to Robert F. Spielhagen**

*We would like to take this opportunity to thank Robert F. Spielhagen for his helpful comments on our manuscript. Below we provide a detailed response to his comments (in italics), indicating the changes that have been made. Line numbers refer to those of the revised manuscript that includes all tracked changes.*

*With kind regards,*

*Mattia Greco (on the behalf of all co-authors)*

General comments

Planktic foraminifera of the species Neogloboquadrina pachyderma are a major carrier of paleoenvironmental information in Arctic and sub-Arctic marine sediments and widely used to reconstruct properties of the upper water column, namely sea-ice coverage, salinity, and temperature. The variability of their geochemical composition (e.g., stable isotopes, Mg/Ca) can be very large within an investigated sediment core. In particular to interpret this variability it is important to know which factors may determine the habitat depth of the species. A number of studies have determined the depth distribution of N. pachyderma under a large variety of boundary conditions (oceanic parameters). A synoptic study involving all the available depth distribution data plus oceanic data from the same sites was much needed but still lacking. The manuscript by Greco et al. fills this gap with a statistical evaluation of mostly published data from the Arctic Ocean and its neighboring seas, amended by some new data from Baffin Bay and results from a numerical model. The combination of biological, physical oceanographic, biochemistry and modelling data results in a novel approach to determine the habitat depth of N. pachyderma on a larger scale and is thematically well suited for the journal Biogeosciences. It is well-written in very good English. The structure of the manuscript follows standard principles of scientific publications. The abstract gives a good overview of the topic, the methodological approach, the major results and the main conclusions. The Introduction chapter gives a good overview of the present knowledge and thereby manifests the problem of defining the factor(s) determining the depth habitat of N. pachyderma. It also describes the general approach applied here and which environmental factors are considered as potentially responsible for the variability in habitat depth. The Material and Methods chapter describes in detail the origin of the data sets used for the following evaluation, the methods to obtain the new data from Baffin Bay, and the statistical methods applied to evaluate and weight the environmental factors determining the habitat depth. I am not an expert on such statistical methods and can therefore not evaluate whether proper attention has been paid to significance levels. The Results chapter lists briefly but in sufficient detail the major outcome of the statistical evaluations, in particular the correlation of habitat depth to individual and combined environmental parameters and how the results from statistical evaluations compare to the model results. The Discussion chapter puts the results of statistical evaluations in context and elaborates which environmental factors are determining the habitat depth. The outcome is discussed with respect to previous hypotheses on which parameters have forced N. pachyderma to live shallower or deeper in the water column. Interestingly, some of these published hypotheses (which in most cases were based on regional studies) are not supported by the conclusions of Greco et al.. The very large data base of the present study (I cannot see that relevant published data sets were left out) is the advantage of the present study and adds significantly to the credibility of the conclusions presented here. To me, the discussion appears to the point and overall sound, and I cannot see that systematic errors may bias the conclusions. These are compiled in the Conclusion chapter which lists the major findings but also open questions which may trigger further research in this field. The figures and tables are mostly clear and easy to understand (see comments below for minor exceptions). Overall, I think this paper is already in a mature state and does not need significant changes. Publication in Biogeosciences is recommended after some minor revision in response to the points listed below.

Specific comments to the authors

Title and manuscript text: Regarding the use of "planktonic" (instead of "planktic") in this manuscript I suggest to read the advice of the Godfather of Paleoceanography, Cesare Emiliani, which can be found here: https://www.cambridge.org/core/journals/journalof-paleontology/article/plankticplanktonic-nekticnektonicbenthicbenthonic/CDF06242F0F9130B7A5A082DFDDFC425

*We agree with the reviewer that the correct expression would be "planktic" as explained in the referred paper. However, the expression "planktonic foraminifera" is more common in the literature than "planktic foraminifera" (up to about 5 times more- as a quick search in Scopus revealed). For practical reasons we therefore prefer to keep using the term "planktonic".*

You use both "habitat depth" and "depth habitat" in the manuscript. The latter is defined on page 5 (line 17ff), the first not. Do these terms have different meanings? If yes, you should explain this. I note that even at the end of the manuscript, in the Conclusion chapter, you still use both terms (page 10, lines 17/18). That is confusing!

*Reviewer 1 also pointed out that we used these two terms interchangeably in the manuscript. We will correct and homogenize the terminology adopting only the term "depth habitat".*

It will be helpful for the reader to receive a bit more information on the PLAFOM2.0 model. As it stands, we just learn that it can predict the seasonal and vertical habitat of N. pachyderma. For those readers who have not studied the Kretschmer et al. (2018) paper in detail, you should use 2-3 lines to explain what the model is based on and which boundary conditions are used.

*We added the following information about PLAFOM2.0 in the text:*

*'This model is driven by temperature, food concentration, and light availability (which matters only for species with symbionts). The species-specific food concentrations are simulated by the Community Earth System Model, version 1.2.2 (CESM1.2, Hurrell et al., 2013) at every time step and are subsequently used by PLAFOM2.0 to calculate the monthly carbon concentration of N. pachyderma and four other species of planktonic foraminifera.'*

Many readers may not be acquainted with all the statistical parameters applied to determine correlations, anticorrelations, significance limits etc.. Those terms used widely throughout the manuscript (e.g., r, p, R, F-test, t-test) should be explained in the manuscript, including a comment on what higher or lower values mean.

*We will add some explanatory notes at page 6 lines 14-16. However, these are all standard statistical concepts that we feel should be familiar to readership of Biogeosciences and we have hence not provided a detailed explanation.*

I suggest not to mix British and American spelling. Either use "paleo" and "...ize" or "palaeo" and "...ise". Please check the entire manuscript for other language cases (e.g., "metres" vs. "meters").

*We will check the manuscript and correct the language inconsistencies.*

Comments by page and line numbers (page/line)

2/1: Better write "dominant plankt(on)ic foraminifer species"

*Done*

2/8: Arctic and North Atlantic oceans

*Done.*

2/10: Here and in other places you mention "chlorophyll a", later you also just write "chlorophyll". Be precise in what you mean.

*We re-checked the usage of the two expressions and specified the data we refer to in the method section.*

2/23-24: "When the organism dies, its calcite shells sink to the seafloor and when preserved in the sediments, they serve ..." Do not mix singular and plural.

*Corrected.*

3/10-14: You are discussing the issue of diel vertical migration again on page 7, lines 18ff, largely repeating what is said here. I suggest to shorten this part in the introduction and put the discussion where it belongs.

*We think that this part is important to highlight why there is still a need for more analyses on the influence of DVM on the DH of N. pachyderma even in the presence of previous investigations.*

3/22: drivers

*Done.*

3/22-26: Very long sentence, hard to read. Split it into two.

*Done.*

4/17: with a conductivity

*Done.*

4/18: obtain vertical profiles

*Done.*

4/19: for all stations

*Done.*

4/20: chlorophyll a concentration profiles

*Done.*

4/21: from the PANGAEA

*Done.*

5/1: all stations

*Done.*

5/2: time of collection

*Done.*

5/11: related to SST

*Done.*

6/23: neither in the complete data

*Done.*

8/21: relationship between DH and environmental parameters

*Done.*

8/21-23: Three sentences starting with "This...". Maybe rephrase?

*Done.*

8/25ff: Better write "In the model, this overestimation of the MLD affects..."

*Done.*

8/34: matter

*Done.*

9/1: depth of

*Done.*

9/2 sea surface

*Done.*

9/6: tolerance limit

*Done.*

9/5-7: Split long sentence into two.

*Done.*

10/2-3: ...evidence ... indicates

*Done.*

10/6-7: compromise between ... and ...

*Done.*

10/22-25: Split long sentence into two.

*Done.*

10/27: mismatch in the

*Done.*

11/6: gratefully acknowledged

*Done.*

12/3-4: Delete blanks!

*Done.*

12/10: ocean

*Done.*

12/13: Carstens, J.

*Done.*

12/13: Sarnthein

*Done.*

13/4: Delete "(Ehrenberg 1861)"

*Done.*

16: A word is missing in the table caption!

*Done.*

Fig. 2c/d: Several symbols are hidden. Possibly use open symbols with no filling?

*We have tried this solution, but even though some symbols overlap we think that the original figure with transparent symbols is clearest and prefer to keep it. It is important to note that the purpose of the figure is not primarily to show each individual point, but the overall absence of a linear trend between sea ice concentration and sea surface temperature, which allows us to investigate their effect on DH independently.*

Fig. 9: Why is "Productivity" related to a filled symbol in the legend while the triangle is open in the figure?

*Corrected.*

---

## Author Comment (AC3) · 9 May 2019

**Response to Antje Voelker**

*We would like to thank Antje Voelker for her helpful comments on our manuscript. Below we provide a detailed response to her comments (in italics), indicating the changes that have been made. Line numbers refer to those of the revised manuscript that includes all tracked changes.*

*With kind regards,*

*Mattia Greco (on the behalf of all co-authors)*

Greco and co-authors compiled new and published vertical abundance data from multi-net tows to evaluate – using statistical approaches – the habitat depth of polar foraminifera N. pachyderma and its relationship to environmental parameters. The study provides new and important insights into a species widely used in paleoceanographic reconstructions, but still with limited information on its living conditions. The authors compare their evidence also to the outcome of the PLAFOM2.0 model (with limited success). With the environmental changes currently occurring in the subpolar North Atlantic and Arctic Ocean this study is for sure timely and relevant for any future studies. The manuscript is well written, the data well presented and deserves to be published in Biogeosciences after minor revision. The following are more general comments that might help improve the manuscript, but are not essential for accepting the manuscript:

1) There exists a very nice study (PhD thesis) [in German] on "The planktonic foraminifera Neogloboquadrina pachyderma (Ehrenberg) in the Weddell Sea, Antarctica" by Doris Berberich published as Berichte zur Polarforschung 195, in 1996. Although this is a different genotype than in the northern hemisphere, it seems that some aspects of the Greco et al. and Berberich observations are similar. So I urge the authors to have a look at this work. I do not know, if the authors could verify with their data is the deeper depth habitat in their data is also related to more adult/ terminal stage specimens and thus potentially to the reproduction cycle. Berberich is also discussing influence of phytoplankton abundance (i.e., food supply) on the foraminifera abundance and sees similar changes in depth as discussed on p. 9 lines 17 to 30. She is referring to Arikawa (1983) when discussing the relationship between N. pachyderma abundance and the deep chlorophyll a maximum. So the Arikawa study is another one the current authors should look into as support for their observation that the depth habitat of their genotype of N. pachyderma appears to be below the chlorophyll maximum. Arikawa, R. (1983), Distribution and taxonomy of Globigerina pachyderma (Ehrenberg) off the Sanriku Coast, Northeast Honshu, Japan. Tohoku Euniv. Sci. Repts., Ser. 2 (Geol.), 53, p. 103-157

*Thank you for these suggestions. Our dataset does not allow for an extensive investigation of differences in DH among size classes of N. pachyderma. For the limited number of stations where we have size data, we found that specimens of smaller size show a deeper depth habitat than the bigger ones. Our subset thus shows the opposite pattern reported by Berberich's. Whether this pattern is real or not requires more detailed size distribution data that we currently do not have.*

*We also thank the reviewer to point us at the interesting work of Arikawa. In his paper, he describes ecological and taxonomical features of the pacific genotype of N. pachyderma (Type VII) collected with horizontal tows from two stations off the Sanriku Coast (Japan). At page 113, lines 1 to 9, Arikawa discusses the general distribution pattern of the planktonic foraminifera concentrations:*

*'At each station, the population density of each species in water columns broadly corresponds to the value of chlorophyll a, which indicates a standing crop of phytoplankton, producer. At both stations, the shallower maximum exists at a depth of 50m and 75m (around or just below the maximum of chlorophyll a concentration).'*

*Our data on type I shows a different pattern and indicates no link between DH and the DCM (Figs 5a and 9). These contrasting observations suggest distinct ecological patterns among the different*

*genotypes of N. pachyderma. However, our study aims to explore the DH of genotype I and we think that a detailed comparison with all other genotypes would require more data and we hence refrain from a rather ad-hoc comparison. We consider this a potential avenue for future research.*

2) p. 4 line 29: did the authors inquire at the AWI oceanography group if the CTD data collected during the ARK campaigns might have been stored there? Since I participated in ARK-X/2, I verified the cruise report and it clearly says on page 95 that at most stations with plankton sampling hydrographic information was obtained with a CTD probe.

*We queried PANGAEA data repository to retrieve the CTD data from the ARK X-2 station and did not find the data for this station. However, we used the counts from that station to investigate the effects of lunar day and DVM on the DH of N. pachyderma.*

More detailed comments to the manuscript itself:

1) throughout the manuscript you are referring to the North Atlantic, even though your samples are actually limited to the subpolar and polar regions of the North Atlantic. If you do not want to use the term Nordic Seas (for the area between Iceland, Greenland, Norway and Svalbard), you could use "northern North Atlantic" to better describe the geographical range of your samples.

*We follow the suggestion and will refer to our sampling area using more precise geographical ranges.*

2) p. 3 line 26: why is food source/supply not mentioned here -although one could argue that this could be a consequence of the change in the environmental conditions?

*We mention the food source at page 3 line 23 where we introduce the environmental drivers of the depth habitat of N. pachyderma.*

3) Material: please provide a table with the stations, date/ year of collection, data source for published data. From your figures one can deduce the season etc., but not how the samples are distributed over the years. Please also provide the name of the station excluded from the Jensen (1998) data set.

*Reviewer 1 also suggested to implement the table in the manuscript. However, due to the size of such a table we prefer to make it available as supplementary material at zenodo.org, where long-term storage is guaranteed. We would like to point out that the link to the table with the complete metadata and environmental data was already provided in the "Data availability" section in the original manuscript. .*

4) p. 4 line 18: please provide the depth until which pigment concentrations were measured. Were the profiles also done down to 300 m?

*We will provide the range of the measurement of the pigment concentrations. The submersible fluorospectrometer recorded data from the surface until 300 m.*

5) p. 5 line 11: small English correction; it should say "related to"

*Done.*

6) p. 7 line 15: it would be good if you could provide the reader with the information how and in which geographical resolution sea ice and chlorophyll are presented in the earth system model, from which PLAFOM2.0 derives its environmental conditions. I wonder if the poor relationship between observations and model might be a resolution problem or sea ice itself not being presented in the model.

*The disagreement is not due to sea ice not being modelled. The here analysed model simulation was an ocean-ice-only simulation of CESM1.2 with active ocean biogeochemistry, whereby the ocean model was coupled to the sea ice model. However, the reviewer rightly points out that the coarse resolution of climate models is often a challenge in model-data comparison. Here, both the ocean*

*component and the sea ice component of CESM1.2 have a zonal resolution of 1° and an increased meridional resolution of ~0.3° near the Equator. However, here we avoid this complication of the relatively coarse resolution of the model by simply looking at modelled versus observed relationships between DH and environmental variables. We will add the following explanation to the text:*

*'By comparing modelled with observed ecological patterns, rather than individual observations, we ensure a more consistent evaluation of the model performance.'*

*We will also make it more explicit that Fig.8 compares modelled DH with modelled sea ice and chlorophyll a concentration. The comparison is thus entirely in model space and not a strict data-model comparison.*

7) p. 9 line 14: if the authors would like to include a study more concentrated on isotopic evidence from the Arctic Ocean they could add the following reference: could also look into Hillaire-Marcel, C., 2011. Foraminifera isotopic records: : : with special attention to high northern latitudes and the impact of sea-ice distillation processes. IOP Conference Series: Earth and Environmental Science 14, doi:10.1088/1755-1315/14/1/012009

*Done.*

8) p. 9 line 30: although the authors write on p 10 line 18 that the species is likely not grazing on fresh phytoplankton, I wonder if type of food source might not be a driver with a preference for "fresh food" during period with a shallower DH and more refracted organic matter during periods when the species prefers the depths below the chlorophyll maximum.

*This is an interesting point. In our data, we observe that the depth habitat is in virtually all cases located below the chlorophyll maximum (Fig. 9), thus N. pachyderma consistently feeds below the DCM, where fresh phytoplankton is rare. Hence, we do not observe indications for food source as a driver of DH. Assuming that N. pachyderma has specific food preferences, such proposed change in its diet also seems unlikely. However, we agree with the reviewer that more investigations on the diet of this species are needed for a better understanding of its ecology.*

---

## Author Comment (AC4) · 9 May 2019

**Response to Caterina Bergami**

*We would like to take this opportunity to thank Caterina Bergami for her helpful comments on our manuscript. Below we provide a detailed response to her comments (in italics), indicating the changes that have been made. Line numbers refer to those of the revised manuscript that includes all tracked changes.*

*With kind regards,*

*Mattia Greco (on the behalf of all co-authors)*

The paper provides interesting paired planktonic foraminfera and environmental data from an important oceanographic region both from new and yet published data in order to better understand the habitat depth preferences of N. pachyderma. This type of studies can help to better understand in which way environmental parameters control the habitat depth and behaviour of this important planktonic calcifier and has an impact on future palaeoecological and palaeoceanographic studies in this area and in other high latitude environments. The authors also compare their evidence to the outcome of the PLAFOM2.0 model, with limited results. The manuscript is well structured and the data well presented and relevant for future studies on the same issues. The amount of figures and tables is adequate to illustrate the results discussed in this paper. The paper deserves to be published in Biogeoscience after some minor revisions that I listed below and in the attached pdf file.

I also suggest to the author a further check of the English language.

Technical points:

- please check the use of the acronyms along the text. Once you define them the first time, please use the same along the text (check in particular DVM, SST and SSS).

*We will make corrections in the text to ensure the correct use of the acronyms once introduced in the text.*

- choose between the term "habitat depth" and "depth habitat (DH)" as definied in the text and use it accordingly. - please check along the text the use of the term "compilation". I would prefer dataset.

*Reviewer 1 and Robert Spielhagen also pointed out that we used these two terms interchangeably in the manuscript. We will correct and homogenize the terminology adopting only the term "depth habitat".*

*Since our analyses are based on an array of data pulled from different sources (PANGAEA, NSIDC, and digitization of existing data from literature), we believe that the term 'compilation' better describes the nature of our dataset.*

- Material and methods sections (from page 5/line 16 to page 6/line 11: In this part of the text some results are mixed with M&M. Please, check and move the results to the following section.

*This is a valid point. However, we consider that part of the methods section more as an evaluation of the methods employed in the analyses and not results. We therefore prefer to keep it in the methods section.*

- Page 4/line 3-4: "We retained all other profiles, despite the differences in the mesh size, counted size fraction and vertical resolution". This phrase is not clear, please re-phrase. What do you mean?

*We refer to the differences in sampling design (mesh size of the nets and sampled depth intervals) and in size fraction analysed. We re-wrote the sentence to increase the clarity:*

*'We retained all other profiles, despite the differences in the sampling design (mesh size and vertical resolution of the sampled depth intervals,) and in the counted size fraction and vertical resolution.'*

- page 9/line 11-12: "sedimentary and plankton specimens". Do you mean fossils and living specimens?

*We use the terms 'sedimentary' and 'plankton' to refer to the source from where the specimens analysed in the cited studies were collected (sediment/cores, water column/sediment traps-plankton hauls). We feel that the distinction suggested by the reviewer is incorrect for two reasons, i) strictly speaking specimens from recent sediments are not fossil or fossilised and ii) dead specimens can also be collected from the water column. However, the distinction is not entirely necessary here and we will delete it.*

---

## Author Comment (AC5) · 9 May 2019

**Response to Katrine Husum**

*We would like to take this opportunity to thank Katrine Husum for her helpful comments on our manuscript. Below we provide a detailed response to her comments (in italics), indicating the changes that have been made. Line numbers refer to those of the revised manuscript that includes all tracked changes.*

*With kind regards,*

*Mattia Greco (on the behalf of all co-authors)*

General comments

The manuscript BG_2019_79 by Greco et al. study which factors that influence the depth habitat of the planktic foraminifera Neogloboquadrina pachyderma using both published and new data together with a suite of statistical methods. The scope of the study is very timely as the regions where this species dominates are subject to climate-ocean changes, hence in order to evaluate current changes and establish robust natural baseline values a better understanding of this species' depth habitat is necessary. The manuscript is well-written and in an advanced state.

Specific comments

1. Figure 10: Introduce this information and figure early on?

*We understand the comment from the reviewer. However, the structure of the paper builds on previously proposed drivers of DH of N. pachyderma and we started with investigating these variables first. The fact N. pachyderma does not appear to track specific temperature, salinity or density (Fig. 10), shows that its habitat is not controlled directly by these environmental variables, lending support to our proposed model. We therefore prefer to keep the original order in which the information is presented.*

2. It would be beneficial to define what is a good correlation/a correlation/a weak correlation, e.g. what is the difference between the r- values of -0.28; -0.38 and -0.60. I am not an expert on statistics but +/- 0.28 seems like a weak correlation? It would also be good to point out that it seems that the correlation improves with a smaller number of observations (e.g. 21 samples without sea-ice)?

*We agree with the reviewer that our model does not explain all observed variability in DH and have discussed the potential reasons for why this is the case on page 8 lines 7-10.*

*To avoid ambiguity, we will reword cases where we used subjective adjectives to describe the correlation (page 7, lines 27-30).*

*'Contrary to observations, the modelled DH shows the highest correlation with the depth of the mixed layer (r = 0.57, p <0.01). Moreover, the observed relationship between the modelled DH and the modelled sea-ice and chlorophyll concentration is lower and of opposite sign compared to the observations (Figs.8a-b).'*

*With regard to the second point raised, we would like to point out that, contrary to the impression of the reviewer, the correlations actually improve both in strength and in significance when more samples are included in the analyses for the subsets of profiles with and without sea ice. (page 7 lines 27-29, Figs 5c and 5d).*

3. The PLAFOM2.0 model is introduced somewhat superficially; more information would be useful.

*A more detailed description of PLAFOM2.0 was also suggested by Robert Spielhagen, so we integrated the following text in the introduction section:*

*'This model is driven by temperature, food concentration, and light availability (which matters only for species with symbionts). The species-specific food concentrations are simulated by the Community Earth System Model, version 1.2.2 (CESM1.2, Hurrell et al., 2013) at every time step and are subsequently used by PLAFOM2.0 to calculate the monthly carbon concentration of N. pachyderma and four other species of planktonic foraminifera.'*

Technical comments

1. Overall: The use of "planktonic" vs "planktic". Please refer to Emiliani 1991: https://doi.org/10.1016/0377-8398(91)90003-O

*We agree with the reviewer that the correct expression would be "planktic" as explained in the referred paper. However, the expression "planktonic foraminifera" is more common in the literature than "planktic foraminifera" (up to about 5 times more- as a quick search in Scopus revealed). For practical reasons we therefore prefer to keep using the term "planktonic".*

2. Overall: There are many acronyms in the paper. Except for SST, SSS, DVM and DH they do not help reading the paper.

*We will check the use of the acronyms and make sure that each is properly introduced and used consistently in the text.*

3. Page 5, lines 6-15 and lines 20-32 ("Materials and methods"): This seem more like a description of results, which it may benefit to move to the start of "Results".

*Caterina Bergami also suggested this change in her review. However, we consider that part of the methods section more as an evaluation of the methods employed in the analyses and not results. We therefore prefer to keep it in the methods section.*